# The Evolving Role of Dendritic Cells in Atherosclerosis

**DOI:** 10.3390/ijms25042450

**Published:** 2024-02-19

**Authors:** Simone Britsch, Harald Langer, Daniel Duerschmied, Tobias Becher

**Affiliations:** 1Department of Cardiology, Angiology, Haemostaseology and Medical Intensive Care, Centre for Acute Cardiovascular Medicine Mannheim (ZKAM), University Medical Centre Mannheim, Medical Faculty Mannheim, Heidelberg University, 69117 Mannheim, Germany; harald.langer@umm.de (H.L.); daniel.duerschmied@umm.de (D.D.); tobias.becher@umm.de (T.B.); 2German Centre for Cardiovascular Research (DZHK), Partner Site Heidelberg/Mannheim, 13092 Mannheim, Germany; 3European Center for Angioscience (ECAS), Medical Faculty Mannheim, Heidelberg University, 68167 Mannheim, Germany

**Keywords:** atherosclerosis, inflammation, immune cells, dendritic cells, innate immunity, adaptive immunity

## Abstract

Atherosclerosis, a major contributor to cardiovascular morbidity and mortality, is characterized by chronic inflammation of the arterial wall. This inflammatory process is initiated and maintained by both innate and adaptive immunity. Dendritic cells (DCs), which are antigen-presenting cells, play a crucial role in the development of atherosclerosis and consist of various subtypes with distinct functional abilities. Following the recognition and binding of antigens, DCs become potent activators of cellular responses, bridging the innate and adaptive immune systems. The modulation of specific DC subpopulations can have either pro-atherogenic or atheroprotective effects, highlighting the dual pro-inflammatory or tolerogenic roles of DCs. In this work, we provide a comprehensive overview of the evolving roles of DCs and their subtypes in the promotion or limitation of atherosclerosis development. Additionally, we explore antigen pulsing and pharmacological approaches to modulate the function of DCs in the context of atherosclerosis.

## 1. Introduction

Atherosclerosis is a chronic disease that affects the arterial walls and is characterized by the buildup of plaques and the narrowing of arteries [1]. The consequences of atherosclerosis include chronic or acute impairment of blood flow that may lead to cardiovascular events, a significant contributor to morbidity and mortality worldwide [2,3]. In spite of major advances in research, the mechanisms underlying the development and progression of atherosclerosis remain incompletely understood.

In recent years, there has been a growing interest in the role of inflammation in atherosclerosis [4]. It is now widely accepted that chronic inflammation within the arterial wall is a major contributor to the development of atherosclerosis. The accumulation and activation of immune cells and the release of inflammatory mediators promote the initiation and progression to advanced atherosclerotic plaques [5,6]. Inflammation further contributes to plaque destabilization and increases the risk of plaque rupture and subsequent thrombosis [7].

Dendritic cells (DCs) are a subtype of antigen-presenting cells (APCs) that reside in the intimal and adventitial spaces of arteries [8,9]. During atherosclerosis initiation and progression, DCs accumulate and orchestrate the immune response by connecting innate and adaptive immunity [10]. Given the major role of inflammation in atherosclerosis, there is increasing interest in developing therapeutic modalities that modulate the underlying inflammatory pathways and mechanisms [11,12]. As the role of DCs in atherosclerosis is evolving and with the advance of innovative therapeutic modalities, modulating DC function may present a novel treatment target. In this review, we outline the function of DCs and specific subtypes and their role in atherosclerosis initiation, progression, and the development of plaque instability. In addition, we highlight DC pulsing as an emerging therapeutic strategy to modify DC function treatment and summarize the effects of available drugs on DC function.

## 2. Cellular Immunity and Atherosclerosis

Cellular immunity, also known as cell-mediated immunity, is one of the two major branches of the immune system and plays a pivotal role in atherosclerosis initiation, progression as well as development of plaque instability. In comparison to humoral immunity (which mainly relies on antibodies), cellular immunity primarily relies on the action of specialized immune cells and cytokines. Its main effectors are T cells, cytokines, and APCs [13].

T cells can be classified into different subtypes, including helper T cells (T_H_; cluster of differentiation 4 positive (CD4+), cytotoxic T cells (CD8+) and regulatory T cells (T_reg;_ CD4+CD25+Foxp3+). In the last decades, T cells have been established as critical drivers and modifiers of atherosclerosis development and progression [1]. T cells drive immune responses to peptide epitopes related to atherosclerosis that are presented by APCs through major histocompatibility complex (MHC) molecules [14,15,16,17]. After recognition through a T cell receptor, naive CD4+ cells differentiate and migrate to the atherosclerotic lesion, mediated by chemokines and their corresponding homing receptors like the CXC chemokine receptor (CXCR) 3, CXCR6, the C-C motif chemokine receptor (CCR) 5 and CCR1 [1,15,18]. Depending on their subtype, T cells may mediate atheroprotective (e.g., T_reg_ cells) or pro-atherogenic effects (e.g., T_H_1) [1,18,19]. Table 1 summarizes T cells and their subtypes that are relevant for atherosclerosis.

Cytokines, such as interleukins (ILs) and interferons (IFNs), are small signaling proteins that coordinate the crosstalk between different cell types. In the setting of atherosclerosis, a variety of cell types that modulate the disease produce and secrete cytokines that are associated with either atherogenic or atheroprotective effects. An increase in pro-inflammatory cytokines, including IFN-y, IL-1β, and tumor necrosis factor (TNF)-α, is related to atherosclerosis promotion and progression, while IL-10, IL-33 and transforming growth factor (TGF)-β are associated with atheroprotective properties [27,28]. During the early stages of atherosclerosis, cytokines contribute to endothelial dysfunction through the activation of endothelial cells. During the later stages of atherosclerosis, cytokines promote the proliferation of smooth muscle cells, apoptosis, and plaque destabilization [29]. An overview of cytokines that are involved in T cell differentiation and activation and of those that are secreted by T cells is provided in Table 1, while Table 2 provides a summary of interferons, interleukins and chemokines and their role in atherosclerosis.

Lastly, APCs are responsible for capturing and processing antigens. Both modified endogenous antigens and, to a lesser extent, exogenous antigens have been implicated in atherosclerosis [33]. APCs can be broadly differentiated into professional APCs (DCs, macrophages and B cells) and non-professional APCs. The defining features of professional APCs include their ability to acquire exogenous antigens and present antigen peptides via MHCII, express co-stimulatory molecules, and secrete cytokines. These features enable professional APCs to train and activate CD4+ T cells. On the other hand, non-professional APCs possess the ability to present peptides derived from endogenous proteins via MHCI to CD8+ cytotoxic T cells, and all nucleated cells can be considered non-professional APCs.

## 3. Dendritic Cells

### 3.1. Discovery and Background

Ralph Steinmann and Zanvil A. Cohn first described DCs in 1973 [34]. Their description was that of a rare subpopulation of adherent cells isolated from mouse peripheral lymphoid tissue with dendritic protrusion. Compared to macrophages, DCs are distinct in that they are more mobile and less phagocytic. In addition, DCs express abundant MHC II proteins on their surface, are potent activators of naive T cells and B cells, and thus play a pivotal role in connecting innate and adaptive immunity [35,36]. While initial experiments were conducted in rats and mice, it soon became apparent that DCs also circulate in human peripheral blood and have similar cytological features compared to rodent DCs when using light and electron microscopy [37].

DCs can be categorized into conventional DCs (cDCs; either type 1 (cDC1s) or type 2 (cDC2s)) and plasmacytoid DCs (pDCs). In mice, all three DC subtypes (cDC1s, cDC2s, and pDCs) originate in the bone marrow in a step-wise manner from pluripotent hematopoietic stem cells (HSCs), via HSC-derived multipotent progenitors (MPPs), common myeloid progenitors (CMPs), monocyte–DC progenitors (MDPs), and common DC progenitors (CDPs) [38]. CDPs give rise to pre-DCs that leave the bone marrow to briefly circulate and to then migrate to peripheral lymphoid and non-lymphoid tissues [39,40]. In addition, pre-pDCs can also arise from common lymphoid progenitors (CLPs) that derive from a different MPP lineage via B-cell-biased lymphoid progenitors (BLPs) [41]. Although counterparts between mouse and human pre-cursors have been established based on surface marker expression, the ontogeny of DCs is less established and still evolving in humans [42]. Figure 1 illustrates the development of cDC1s, cDC2s and pDCs in mice starting from HSCs.

The identification of DC subsets in different tissues, their respective lineage, and different states of activation have been a major focus of research during the last decades. Based on this research and work on macrophages and monocytes, the nomenclature of DCs as well as the entire mononuclear phagocyte system (MPS) has undergone continuous changes. Guilliams et al. recently proposed a unified, sequential nomenclature based on ontogeny (level one nomenclature) and function, location, and/or phenotype (level two nomenclature) for the MPS that encompasses DCs, monocytes, and macrophages [43]. Accordingly, cDC1s can be differentiated by their dependence on transcription factors (TFs) IRF8, BATF3, ID2 and NFIL3 from cDC2s (IRF4-dependent) and pDCs (TCF4-dependent) [42,43,44]. In humans, the expression of lineage-defining TFs for cDC1s is conserved, while there are significant differences when comparing mouse and human cDC2 and pDC lineage-determining TFs [42].

Expression markers that provide additional information about cell identity, functional specialization, or cell localization further supplement the ontogeny-based approach [45,46,47]. In general, all DCs express varying levels of MHCII and CD11c [48]. In mice, the cDC1 subset can be identified by assessing the surface marker expression of CD103, XCR1, CD8α or CLEC9A, while the cDC2 subset expresses CD11b, CD41, and CD172a, and pDCs are characterized by B220, CD45RA, SiglecH, Ly-6C, and BST2 [48,49]. In humans, different surface markers characterize the DC subtypes. While cDC1s are identified by the expression of CD141, CLEC9A, XCR1, and CADM1, cDC2s express CD1C, CD172A, and FcεR1A. PDCs can be identified through CD123, CD2, and CD45RA [49]. In addition, DCs express multiple chemokine receptors and ligands (CCLs) that are important for DC migration [50]. Maturation of DCs can be monitored by assessing the expression of the co-stimulatory molecules CD80, CD86, and MHCII. Depending on the respective subset, tissue location, and activation stage, additional markers may be required to identify DC subsets of interest (e.g., splenic CD8α + DCs). Transcription factors, expression markers and activation markers that distinguish cDC1s, cDC2s and pDCs are summarized in Figure 1.

With the increasing use of single-cell technologies and the identification of novel DC subsets, a continuing adaption of DC nomenclature and classification DC subsets can be expected [47,51,52]. The continuing evolution of our understanding of DC subtype identity and ontogeny in combination with the observation that a variety of immune cells may express similar surface markers (e.g., CD11c in DCs and macrophages) complicates attributing past study results to single DC subtypes.

DC migration, characterized as the ability of DCs to reach target locations within the body, is a pivotal step for the initiation of pro-inflammatory or tolerogenic immune responses. DCs first enter the blood stream in the precursor stages, as outlined above, and are then distributed to lymphoid and non-lymphoid tissues [39]. In non-lymphoid tissues, immature DCs undergo a spontaneous maturation program to become semi-mature DCs or mature DCs upon encountering and interacting with antigens. After upregulation of CCR7, both semi-mature and mature DCs migrate towards terminal lympathic vessels that express the CCR7 ligand CCL21. After entering the lymphatic vasculature, DCs are passively transported towards the draining lymph node, where they migrate mainly towards the T cell zone, a prerequisite for an efficient interaction with T cells [39]. Migratory subsets of DCs can be distinguished based on surface markers and react to different migration cues.

CDCs can be characterized as resident sentinel cells that acquire and internalize antigens to maintain immune homeostasis. Subsequent activation of cDCs is characterized by the ability to interact with immune cells, which mainly include T cells but also B cells and nonimmune cells. DCs possess pattern recognition receptors (PRRs; including toll-like receptors (TLR) and c-type lectin receptors (CLRs)) that recognize a variety of pathogen-associated molecular patterns (PAMPs) and damage-associated molecular patterns (DAMPs), a process that leads to DC activation and maturation. Activated cDC1s present acquired antigens in a MHCI-dependent process to CD8+ T-cells and are specialized in CD4+ T_H_1 polarization against intracellular pathogens and tumors [53,54]. Activated cDC2s, on the other hand, preferentially prime CD4+ T cells for T_H_2 or T_H_17 polarization [10,55,56]. Lastly, pDCs are specialized in recognizing DNA and RNA, either viral, microbial, or self. This process involves TLRs like TLR 7 and TLR 9 and results in the production of high amounts of type I and III IFN [41]. Besides orchestrating the immune response to a threat, DCs also contribute to immune tolerance. These tolerogenic properties are conveyed through clonal anergy and deletion, the metabolic modulation of T cells, and the production and secretion of anti-inflammatory cytokines [57]. These dual roles explain why the modulation of DC function may result in net deleterious or beneficial results on atherosclerosis.

### 3.2. Role and Function of Dendritic Cells in the Vasculature

DCs have been reported as early as 1995 in ultrathin sections of human aortic intima, collected from trauma victims during autopsy and examined by electron microscopy and immunhistochemistry [58]. In this study, Bobryshev and Lord described cells with a DC-like appearance mainly located in the sub-endothelial space that were termed vascular DCs. This observation was later confirmed and extended to other vascular beds (e.g., carotid, iliac, mesenteric, subclavian, and temporal arteries) [8,17,59,60,61,62]. Vascular DCs were also reported in the arteries of other species (e.g., rabbits and mice), but not in veins [9,62]. In C57BL/6 mice, in the absence of an atherosclerosis-promoting genetic background or diet, DCs in the vascular tissues consist of the main DC subsets (cDC1s, cDC2s, and pDCS) and are characterized by an immature, more tolerogenic phenotype compared to splenic DCs [63].

Interestingly, DCs in the vasculature seem to accumulate in areas that are more prone to atherosclerotic plaque formation, such as branch points, curvatures, or areas with increased turbulent flow, providing the first implication of a potential involvement in atherosclerosis initiation and development. This notion is further supported by the observation that the abundance of subendothelial DCs correlates with the vulnerability for plaque development across several mouse strains, and that subendothelial DCs are absent in areas that are protected from atherosclerosis [62]. Aging, a non-modifiable risk factor for atherosclerosis development, is also associated with the accumulation of CD11c+ cells in mouse aortas [64]. In healthy mouse aortas, CD11c+ DCs can effectively cross-present antigens on MHCI to CD8+ T cells, indicating a role in maintaining immunogenic hemostasis [17].

### 3.3. Role and Function of Dendritic Cells in Atherosclerosis

Compared to the low numbers of DCs in healthy vasculature, DCs increase in abundance during plaque development in mice and humans [14,58,65,66,67,68,69,70,71]. Table 3 provides an overview of DC localization, identification markers and their change in atherosclerosis in mice and humans.

In human carotid artery plaques, DCs form focal clusters and exhibit different distribution patterns in initial lesions (more common in the intima) compared to fully formed plaques (mainly in the plaque shoulder and marginal parts of the plaque core, with fewer DCs in the central part) [65]. DCs are also more prevalent in the plaque shoulder and core in unstable plaques compared to stable ones. As mentioned above, DCs are primarily found in the intima, acting as sentinel cells that probe for antigens. In addition, DCs seem to have the ability to extend cellular processes into the vessel lumen, indicating that vascular DCs may also probe for antigens in close proximity to the endothelial barrier [17].

In atherosclerosis progression in mice, DCs have the ability to accumulate intracellular lipids and transform into CD11c+ foam cells [73]. These activated DCs interact with CD4+ T cells within the aortic wall, triggering T cell activation, proliferation, and the release of pro-atherogenic cytokines such as IFN-γ and TNF-α [14]. Outside of atherosclerotic lesions, DCs have also been observed in adventitial artery tertiary lymphoid organs (ATLOs) that form in response to smooth muscle cell (SMC) expression of lymphorganogenic chemokines in areas where SMCs overlay intimal plaques [74,75]. This suggests that, apart from atherosclerotic plaques and secondary lymphoid organs/tissues, ATLOs may also facilitate DC-T cell interactions and cross-talk [76].

The activation and function DCs are closely linked to lipid metabolism, suggesting that disturbances in circulating lipids, as seen in many patients with atherosclerosis, may impact DC function. Interestingly, DCs, however, appear to maintain their capacity to prime CD4+ T cells and regulate their response in experimental atherosclerosis in mice, even in the presence of hypercholesterolemia [77]. On the other hand, there is evidence to suggest that DCs themselves can influence systemic lipid levels. Several studies that have manipulated DC function through genetic modifications have reported changes in circulating lipid levels, implying a role for DCs in systemic lipid metabolism [78,79,80,81,82,83,84]. Additionally, it has been observed that CD11c+ DCs are capable of producing apolipoprotein E (ApoE) [85]. Although the precise mechanisms are not yet fully understood, the available evidence supports a connection between DCs and systemic lipid metabolism in the context of atherosclerosis.

DC activation after antigen recognition and opsonization is pivotal for effective T cell priming, and defective DC activation and maturation have been linked to atherosclerosis exacerbation [80,86,87]. The ability to increase MHCII expression during maturation is a hallmark of DCs and is required for antigen presentation to CD4+ T cells. Depending on costimulatory molecule expression, activated CD4+ T cells can differentiate into T_H_ (including pro-atherogenic T_H_1) or atheroprotective T_reg_ cells [88]. MHCII deficiency affecting all APCs results in a lack of CD4+ T cells in the spleen and an increase in CD8+ T cells, as well as a reduction in T_reg_ cells and plasma IL-10 levels [80]. Subsequently, MHCII ablation in *Apoe* knockout mice (*Apoe*−/−) aggravates atherosclerosis, indicating an important atheroprotective role of MHCII-dependent antigen presentation [80]. Mechanistic studies implied that despite reduced inflammatory activity, both systemically and locally, in the artery wall, lower plasma lipids, and fewer monocytes in the spleen, the decreased T_reg_ cell count contributes mainly to increased plaque formation in MHCII-deficient mice.

#### 3.3.1. CD11c+ Dendritic Cells

Several studies have used the pan-dendritic marker CD11c to genetically modify the DC population and investigate its impact on atherosclerosis [86,87]. However, it is important to acknowledge that this approach has certain limitations. Surface marker CD11c lacks specificity and can introduce unintended effects, as it is also expressed by macrophages and activated T cells during inflammation [89]. Moreover, relying solely on CD11c expression does not enable the study of individual subpopulations within the DC population. Various cellular pathways in DCs have been manipulated in previous studies to explore their role in atherosclerosis.

##### Krüppel-like Factor 2

Transcription factor Krüppel-like factor 2 (KLF2) has been implicated in the regulation of DC activation and T cell responses. The deficiency of KLF2 has been associated with the upregulation of DC maturation markers, such as CD40 and CD86 [86]. Consequently, DC activation leads to lymphopenia due to increased T cell turnover. The elevated activation of CD4+ and CD8+ T cells, along with the secretion of pro-atherogenic cytokines like IL-2, IFN-γ, and TNF-α, exacerbates atherosclerosis in mice lacking the low-density lipoprotein receptor (*Ldlr* knockout mice) and receiving KLF2-deficient DCs. These findings suggest a potential atheroprotective role for KLF2 in DCs.

##### Myeloid Differentiation Primary Response Protein 88

Myeloid differentiation primary response protein 88 (MYD88) is a cytoplasmic adapter protein that plays a crucial role in the signaling pathways of toll-like receptors (TLRs) and interleukin-1 receptors (IL-1Rs), facilitating the transmission of extracellular signals (such PAMPs and DAMPs) into intracellular activation of signaling cascades, including nuclear factor kappa-light-chain enhancer of activated B cells (NF-κB) [90]. In *Ldlr*−/− mice, selective depletion of Myd88 in CD11c-expressing cells (DCs) results in defective DC maturation, leading to reduced T cell activation. This depletion also leads to an increased number of CD11c+ cells and monocytes in atherosclerotic lesions, while the population of atheroprotective T_reg_ cells is decreased [87]. Consequently, CD11c-selective depletion of Myd88 exacerbates atherosclerosis, suggesting that the overall effect of MYD88-mediated DC activation and maturation is atheroprotective, primarily through the promotion of atheroprotective T_reg_ cells.

##### Human B-Cell Lymphoma 2

The overexpression of the anti-apoptotic gene human B-cell lymphoma 2 (*hBcl-2)* in CD11c+ cells has been shown to increase the lifespan and immunogenicity of the DC population [78]. This leads to an expanded DC population, which in turn enhances T-cell activation, as evidenced by changes in the expression profiles of T_H_1- and T_H_17-related cytokines. Although these changes in T-cell activation and the promotion of a T_H_1-polarized immune response can be considered atherogenic, it is interesting to note that no acceleration in atherosclerosis development on a cholesterol diet was observed in this model. This might be attributed to a concurrent decrease in plasma total cholesterol, particularly LDL and VLDL cholesterol [78]. Accordingly, diphtheria toxine-induced CD11c+ cell depletion resulted in a transient increase in plasma cholesterol levels, a phenomenon not observed when selectively depleting CD11b+ cells [91].

##### TGF-β Receptor II Signaling

The TGF-β superfamily encompasses three isoforms (TGF-β1, TGF-β2, and TGF-β3) and plays a crucial role in regulating tissue inflammation, repair, remodeling, and fibrosis in cardiovascular diseases [92]. Functional inactivation of TGF-β receptor II signaling specifically in CD11c+ cells was shown to result in aggravated atherosclerosis in the *Apoe*−/− mouse model. This aggravation is likely mediated by an increase in CD4+ and CD8+ T cell counts within atherosclerotic lesions, while T_reg_ cells remain unchanged. The observed worsening of atherosclerosis occurred despite lower cholesterol levels and no changes in triglycerides [79].

##### Hypoxia-Inducible Factor

DCs engage in constant interactions with their microenvironment, influenced by cytokines and metabolites, leading to reciprocal effects. As atherosclerotic plaques progress, hypoxia further impacts cellular processes and metabolites, including DCs. Hypoxia-inducible factor (HIF)-1α, a transcription factor induced under hypoxic conditions, is also upregulated in response to oxidized low-density lipoprotein (oxLDL) exposure [93,94]. Deletion of Hif1α specifically in CD11c-expressing cells (DCs) exacerbates the size of atherosclerotic lesions, modifies plaque characteristics (increased necrotic core area), and enhances the presence of CD3+ cells within plaques. In this model, local and systemic T-cell activation, as well as T_H_1 polarization, further suggest the involvement of HIF-1α in DCs in restraining T-cell activation and T_H_1 polarization during atherosclerosis development. Transplanting bone marrow derived from *Zbtb46*-CRE *Hif1α*-fl/fl double knockout mice into irradiated *Ldlr*−/− mice to generate a DC-specific knockout of Hif-1α leads to increased atherosclerosis. Mechanistically, HIF-1α activation in DCs stimulates fatty acid synthesis and intracellular storage of fatty acids, and it also influences the production of bioactive lipid mediators. Ultimately, these effects may suppress DC activation, resulting in a net atheroprotective effect under hypoxic conditions [95].

##### Indoleamine 2,3-Dioxygenase

In addition to cytokines, DC metabolites also have the ability to modify the plaque microenvironment and influence disease progression. One such metabolite is indoleamine 2,3-dioxygenase (IDO), which is one of the rate-limiting enzymes in the kynurenine pathway, a significant pathway for tryptophan metabolism. IDO catalyzes the oxidation of L-tryptophan to N-formylkynurenine. Metabolites generated in the kynurenine pathway, as well as the expression of IDO, have been implicated in modulating the progression of atherosclerosis, although the overall effects (atheroprotection or progression) are not yet fully understood.

DCs express IDO, and in *Apoe*−/− mice fed a high-fat diet (HFD), IDO-expressing DCs accumulate in atherosclerotic lesions [96]. Stable overexpression of IDO in an immature DC cell line derived from bone marrow progenitor cells, and subsequent transfer into the *Apoe*−/− mouse model reduces atherosclerosis development, independent of changes in lipid metabolism. This effect is accompanied by increased levels of kynurenine, as well as CD4+CD25+Foxp3+ T_reg_ cells and decreased T_H_1 cell numbers in the aorta. These findings suggest that DC-derived kynurenine and its receptors, such as the aryl hydrocarbon receptor, may play a role in T_reg_ cell expansion and mediating atheroprotective effects.

##### Cyclooxygenase 1, 2 and Endothelial Nitric Oxide Synthase

Cyclooxygenase (COX) is the rate-limiting enzyme in the synthesis of prostaglandins (PGs) from arachidonic acid. The COX gene encodes the two isoenzymes COX-1 and COX-2. While COX-1 is constitutively expressed, COX-2 is induced rapidly in response to inflammation-causing cytokines [97]. In the context of atherosclerosis, COX-2 expression is upregulated in endothelial cells, smooth muscle cells and macrophages, and it accumulates in atherosclerotic lesions [98,99]. PGs have a significant impact on atherosclerosis and cardiovascular health and can exert atheroprotective (e.g., PGI_2_) and pro-atherogenic effects (e.g., PGE_2_).

COX-2 expression and PGE_2_ production are increased in DCs after lipopolysaccharide (LPS) exposure [100,101]. COX-2-derived PGE_2_ has been linked to the survival of DC progenitors and the FLT3 ligand-dependent development of CD11c+CD11b+ cDCs (but not pDCs) [102]. Moreover, PGE_2_ signaling modulates DC migration and maturation via E-prostanoid 2/E-prostanoid 4 (EP2/EP4) receptors [103,104,105]. It also affects the release of IL-10, IL-12 and IL-23 by DCs [100]. In a mouse model of type 1 diabetes-related accelerated atherogenesis, however, myeloid cell-targeted deletion of the EP4 receptor impaired PGE2 signaling and altered cytokine production in DCs but did not modulate atherosclerosis development [106].

Nitric oxide (NO) is generated by one of the three NO synthase (NOS) isoforms called neuronal (nNOS; NOS3), inducible (iNOS; NOS2) and endothelial (eNOS; NOS1) NOS [107]. These NOS enzymes facilitate the conversion of L-arginine, NADPH, and O2 into L-citrulline, NADP+, and NO. NO, a gas that can permeate membranes, plays a crucial role in modulating vascular tone, endothelial function, leucocyte chemotaxis, and platelet adhesion, and the disturbances in the equilibrium of NO are significant contributors to atherosclerosis. In murine monocyte-derived DCs, iNOS expression has been observed and can be induced by IFN-γ or LPS [108,109]. NO triggers metabolic reprogramming in DCs, characterized by sustained glycolysis and the suppression of mitochondrial activity, which is mediated by mTOR/HIF1α/iNOS [110]. These metabolic changes are associated with increased production of inflammatory cytokines, upregulation of costimulatory molecules, enhanced T cell stimulatory capacity, and prolonged DC survival [109,111]. Although iNOS has been detected in DCs present in human carotid artery atherosclerosis, the specific effects of NO on DCs in this context remain unclear [112].

The data presented provide compelling evidence for the involvement of the overall CD11c+ DC population in modulating atherosclerosis. It is important to note that different subtypes of DCs may have distinct contributions to plaque development, each potentially contributing to net pro-atherogenic or atheroprotective effects as outlined in Figure 2.

In the following section, we describe in more detail the various DC subtypes and their roles in atherosclerosis. The role of monocyte-derived DCs, characterized by a dependence on granulocyte-macrophage colony-stimulating factor (GM-CSF) and expression of CD11b+CD11c+ MHCII+/− CD64+MerTK+/− CD206+, is not part of this review given the different ontology.

#### 3.3.2. Conventional Dendritic Cell Subtypes 1 and 2

Studies in mice identified cDC1s and cDC2s as distinct subsets of DCs that modulate atherosclerosis development. In terms of relative abundance in the aorta of healthy C57BL6 mice, the cDC2 subpopulation shows the highest percentage, followed by cDC1s and pDCs [72]. In the setting of atherosclerosis in *Apoe*−/− mice, pDCs significantly increase while the subpopulation of cDC2s decreases and cDC1s remain unchanged [72].

To elucidate the role of cDC1s and cDC2s independently from each other, a number of approaches have been applied in mice to modulate the development and maintenance of individual DC subtypes [67,113,114,115,116]. Table 4 summarizes the different approaches and the observed net effect on atherosclerosis development.

##### Fms-like Tyrosine Kinase 3

The Fms-like tyrosine kinase 3 (FLT3) ligand is a cytokine that is required for the development of FLT3-dependent cDCs and pDCs [124]. *Flt3*−/− mice exhibit decreased numbers of pre-DCs, cDCs, and pDCs in lymphoid tissues. Moreover, these mice show a significant reduction in the peripheral cDC1 population, while the cDC2 population remains relatively unaffected [125]. *Ldlr*−/− *flt3*−/− double knockout mice on a HFD showed a significantly increased lesion size despite the absence of changes in plasma lipid levels [67]. Although the total T cell numbers were not altered, there was a significant reduction in T_reg_ cell numbers, along with increased mRNA expression of IFN-γ and TNF-α in the aorta. These findings suggest that cDC1s may have an atheroprotective effect by stimulating T_reg_ cells and modulating inflammatory cytokines.

##### Transcription Factors Basic Leucine Zipper Transcription Factor ATF-like 3 and Interferon Regulatory Factor 8

Several studies have investigated the role of the basic leucine zipper transcription factor ATF-like 3 (BATF3), which is required for the development of lymphoid tissue cDC1s and peripheral cDC1s, in atherosclerosis, yielding different results [113,114,115]. Li et al. reported a significant reduction in atherosclerosis in *Batf3*−/− *Apoe*−/− double-knockout mice, likely attributed to a decrease in pro-atherogenic T_H_1 cells and the pro-inflammatory cytokine IFN-γ [113]. However, this finding was not replicated in *Batf3*−/− *Ldlr*−/− double-knockout mice [115]. Additionally, transplanting the *Batf3*−/− bone marrow into *Ldlr*−/− mice did not impact atherogenesis [114]. These divergent results suggest that the role of cDC1s, regulated by BATF3, in atherosclerosis may be influenced by various factors and require further investigation.

Interferon regulatory factor 8 (IRF8) is a transcription factor that is required for the survival of terminally differentiated cDC1s [126]. Restricted deletion of IRF8 in CD11c+ cells ablates lymphoid and non-lymphoid cDC1s [116]. The absence of cDC1s is associated with a reduction in CD3+ T cell accumulation within atherosclerotic lesions, an overall decrease in atherosclerosis, and alterations in plaque characteristics, such as a smaller acellular area, decreased collagen content, and higher smooth muscle cell content (the latter observed in female mice only, suggesting potential sex differences), in the *Ldlr*−/− mouse model fed an HFD. Additionally, there is a modest increase in plasma total cholesterol levels. These findings indicate that IRF8-dependent cDC1s are involved in promoting pro-atherogenic T-cell responses, including T-cell activation, T-cell-dependent antibody and cytokine production, as well as reduced T follicular helper and germinal center B-cell responses [116].

##### Autophagy-Related 16-like 1

In addition to genetic perturbations of transcription factors involved in cDC1 and cDC2 development, the function of cDC subtypes also plays a crucial role in atherosclerosis development. Specifically, in the context of hypercholesterolemia, vascular DCs exhibit increased autophagic flux compared to macrophages [117]. Disruption of autophagy by deleting autophagy-related 16-like 1 (*Atg16l1*) has been shown to limit the extent of atherosclerosis and influence plaque phenotype, such as reducing the necrotic core area. Interestingly, this pathway seems to have different roles in different DC subtypes, as deletion of *Atg16l1* in CD8α+ DCs did not affect the development of atherosclerosis compared to CD11b+ DCs. The overall atheroprotective effect was mediated by an increase in T_reg_ cells and a reduction in T_H_1 cells within the aorta. These findings highlight the importance of autophagy in regulating the function of cDC subtypes and their impact on atherosclerosis progression.

##### C-Type Lectin Receptors CLEC9A and CLEC4A4

Lastly, PRRs, including TLRs and CLRs play a pivotal role in DC biology [127]. Dendritic cell natural killer lectin group receptor-1 (DNGR-1; also known as CLEC9A) is a CLR predominantly expressed on CD8a+ cDC1s, and it acts as a receptor for dead cell-associated antigens, which may accumulate in the necrotic cores of advanced atherosclerotic lesions [128]. In the absence of DNGR-1, the cross-priming ability of CD8a+ cDC1s to cell-associated antigens is reduced. Notably, in *Apoe*−/− *Clec9a* −/− mice fed an HFD, a significant reduction in atherosclerosis was observed. Mechanistically, CLEC9A on CD8a+ cDC1s regulates the production of the atheroprotective cytokine IL-10 in CD4+ T cells. In the absence of CLEC9a on CD8a+ cDC1s, increased IL-10 expression levels are observed in the spleen, atherosclerotic lesions, and the aorta. These findings highlight the importance of DNGR-1 in regulating the immune responses mediated by CD8a+ cDC1s and its implications in atherosclerosis development.

CLEC4A4, another CLR, is primarily expressed on CD8- cDC2s and has been implicated in mediating pro-atherogenic effects [118]. In *Ldlr*−/− *Clec4a4*−/− double-knockout mice fed a Western-type diet, a reduction in plaque area and necrotic core at the aortic sinus was observed compared to *Ldlr*−/− single-knockout mice. Interestingly, this effect was observed in both male and female mice, while decreased cholesterol and triglyceride levels in combination with decreased circulating neutrophils and inflammatory monocytes were observed in male but not in female mice. This suggests sex hormone-related differences in lipid metabolism and cDC2 function that may contribute to these differences. Previous studies have implicated estrogen and estrogen receptor alpha (ERα) signaling in the development and functional capacities of DCs [129,130]. ERα, a ligand-dependent transcription factor, is expressed in both progenitor and mature DCs [131]. Its signaling promotes IRF4 expression in monocyte-derived DC progenitors and their subsequent differentiation [132]. In the FLT3 ligand-dependent model of DC differentiation, ERα signaling enhances the development of cDCs and pDCs, which exhibit more pronounced pro-inflammatory cytokine production in response to TLR stimulation. The overall effect of estrogen and ERα signaling on cDC2s in particular is currently unclear. Further investigation is also needed to understand the precise role of CLEC4A4 in cDC2s and the molecular mechanisms underlying its suggested pro-atherogenic effect.

#### 3.3.3. The CCL17 Dendritic Cell Subtype

Chemokine CCL17 (also called thymus- and activation-regulated chemokine or TARC) is expressed by a subset of CD11b+CD8− DCs that account for 2-30% of CD11c+ DCs in the thymus, the mesenteric and cutaneous lymph nodes as well as the lungs but are absent in the spleen. This subset is further characterized by the expression of CD40 and CD86 and has the ability to induce T cell proliferation and secretion of IFN-γ and IL-10, indicating a mature DC phenotype [133]. CCL17 expression is upregulated in response to IL-4 and IL-13 treatment, and it can be detected in aortic samples containing fatty streak lesions, calcified type IV atherosclerotic plaques, and plaque material from carotid endarterectomy [134].

In *Apoe*−/− mice fed an HFD, the CCL17-expressing DC subtype is present in atherosclerotic lesions, albeit in low abundance, and can be identified using flow cytometry [119]. Knockout of CCL17 in mice leads to a reduction in atherosclerotic burden and a more stable plaque phenotype. This effect is mediated through CCL17’s role in recruiting CD4+ T cells and limiting the expansion of atheroprotective CD4+CD25+Foxp3+ T_reg_ cells Additionally, blocking CCL17 with an antibody has been shown to significantly reduce lesion formation and potentially increase T_reg_ cell numbers in the aortas of *Apoe*−/− mice fed an HFD. These findings suggest that targeting the CCL17 pathway could be a promising approach for the treatment of atherosclerosis (Table 1) [119].

#### 3.3.4. Plasmacytoid Dendritic Cells

PDCs, also known as plasmacytoid DCs, can be defined as PDCA1+, TCF4+, B220+, or SiglecH+ CD11c+ cells in mouse studies. They represent the smallest subset of DCs. Similar to cDCs, PDCs have been shown to accumulate in murine models of atherosclerosis and can also be identified in human atherosclerotic lesions, often found near the plaque shoulder and areas bordering the necrotic core [120,121,135]. PDCs have been observed to ingest oxidized low-density lipoprotein (oxLDL), resulting in increased expression of CD36 but not CD68, MHCII, or CD86, and without production of IFN-α [121]. OxLDL appears to enhance the phagocytic capabilities of PDCs. Additionally, PDCs are effective in presenting antigens to CD4+ cells through MHCII and induce antigen-specific proliferation of CD4+ T cells, which is potentiated by pretreatment with oxLDL [121,122,123]. Consistent with their canonical function, PDCs in atherosclerotic plaques express IFN-α [135]. IFN-α production is higher in inflamed regions of the plaque compared to non-inflamed areas, and it induces upregulation of TNF-related apoptosis-inducing ligand (TRAIL) on CD4+ T cells, as well as TRAIL-mediated SMC necrosis, suggesting a potential role in plaque destabilization [135]. Several studies have investigated the role of PDCs in atherosclerosis using different approaches, such as genetic ablation or antibody-mediated depletion of PDCs targeting PDCA1, with variable results [120,121,122,123].

In a mouse model utilizing *Cd11c-Cre × Tcf4* (−/flox) bone marrow transplant approach in *Ldlr*−/− mice to selectively ablate pDCs, a significant reduction in pDCs was observed in the blood, spleen, lymph nodes, and aortas [123]. Interestingly, a cDC-like population (CD11c+MHCII+) derived from converted *Tcf4*−/− pDCs was observed, with no significant changes in the ratio of CD11b+ and CD8α+ cells within this population. Selective ablation of pDCs in this mouse model resulted in a slight increase in plasma total cholesterol levels, but more importantly, a significant reduction in atherosclerosis. These findings suggest that pDCs may contribute to the development of atherosclerosis by functioning as APCs that present atherogenesis-specific antigens to IFN-γ–producing CD4+ T cells via MHCII. This highlights the potential role of pDCs in promoting pro-atherogenic immune responses in atherosclerosis.

##### Anti-Mouse PDCA1 Antibodies

In an *Apoe*−/− mouse model, the use of an anti-mouse PDCA1 antibody resulted in the depletion of pDCs in the aorta and spleen, leading to a significant reduction in atherosclerosis formation and a more stable plaque phenotype [122]. With this approach, no significant differences in cholesterol and triglyceride levels were observed between the groups. Concurrently, T-cell activation in the spleen and pro-atherosclerotic plasma cytokines were reduced. A separate study by Döring et al. also utilized the pDC-depleting PDCA1 antibody in *Apoe*−/− mice fed a high-fat diet [121]. This study confirmed the previous findings, demonstrating a reduction in plaque size in the aortic root and aorta upon pDC ablation, accompanied by a significant decrease in IFN-α serum levels.

In a study conducted by Daissormont et al., a different antibody (120G8 mAb) was used to deplete pDCs in the *Lldr*−/− mouse model that was fed a high-fat diet and underwent atherosclerosis induction by placing semiconstrictive collars around the carotid artery [120]. Surprisingly, depletion of pDCs in this study resulted in accelerated atherosclerosis and a more unstable lesion phenotype. This observation was associated with an increase in plasma IFN-gamma levels and increased accumulation of T cells within the lesions in the absence of pDCs. These findings suggest a role for pDCs in mediating CD4+ T cell proliferation, potentially through the immunomodulatory enzyme IDO.

Approaches to activate pDCs have been utilized to gain further insights into their role in atherosclerosis. For example, exposure to CpG or cramp/DNA complexes has been shown to exacerbate atherosclerosis in the *Apoe*−/− mouse model, which aligns with the notion that pDCs, in general, contribute to the development of atherosclerosis [121].

The contrasting findings underline the complexity of pDCs’ role in atherosclerosis and highlight the need to gain a better understanding of pDC biology and their interactions within the atherosclerotic microenvironment.

### 3.4. Dendritic Cells and Therapeutic Opportunities

#### 3.4.1. Dendritic Cell Pulsing

Several exogenous and autoantigens have been implicated in the initiation of an immune response and the induction and progression of atherosclerosis, including cytomegalovirus, hepatitis C virus, human immunodeficiency virus, and others, as well as autoantigens including heat shock proteins and LDL/oxLDL and its major protein apolipoprotein B-100 (ApoB-100) [13,136]. More than 100 ApoB-100 epitopes that are associated with an immune response in humans have been identified, and some have been shown to reduce atherosclerosis when used as part of a vaccination approach [136,137]. Passive (i.e., administration of a preformed antibody directly) and active (i.e., administration of antigens to induce an antibody response directed against factors that contribute to atherosclerosis) vaccination approaches for atherosclerosis have been summarized elsewhere [138].

Given the central role of DCs in innate immunity and in orchestrating adaptive immunity, the question has emerged whether DC function can be modulated to ameliorate atherosclerosis. Pulsing of DCs is a sequential in vitro process in which DCs are first derived from an autologous origin, then exposed to and loaded with antigens, followed by maturation under optimized conditions. Several studies have shown that in vitro incubation of bone marrow-derived DCs with atherosclerosis-associated antigens, followed by infusion, can affect atherosclerosis formation [139,140,141,142].

Pulsing DCs with copper-oxidized LDL has been shown to reduce atherosclerosis and stabilize plaques in *Ldlr*−/− mice. These mice underwent perivascular collar placement around the carotid arteries to accelerate local atherosclerosis [139]. Notably, the presence of IgG antibodies against copper-oxidized LDL indicated a DC-mediated humoral immune response. Furthermore, antigen exposure to splenocytes ex vivo resulted in more pronounced T cell proliferation and reduced production of IFN-γ. Of note, the effects of pulsed DC treatment were most evident in the carotid arteries, while the decrease in atherosclerosis in the aortic root was minor and not statistically significant. This suggests that the effectiveness of the treatment may vary across different regions of the vasculature, potentially influenced by the underlying molecular mechanisms driving atherosclerosis development.

Pulsing bone marrow-derived myeloid DCs with IL-10 and the ApoB-100 antigen also reduced atherosclerotic surface lesion area in *Ldlr*−/− mice expressing full-length human ApoB-100 while inducing a humoral immune response [141]. The pulsed DCs in this study acquired a more tolerogenic phenotype, characterized by the ability to induce antigen-specific T_reg_ cells and to downregulate the pro-inflammatory T-cell response to the loaded antigen.

Pulsing DCs with different antigens has yielded contrasting results [140,142]. Pulsing DCs with malondialdehyde-modified LDL (MDA-LDL) ex vivo aggravated atherosclerotic lesions in the *Apoe*−/− mouse model [140]. Transfer of these pulsed DCs resulted in the production of IgG and IgM antibodies against MDA-LDL and increased secretion of IFN-γ from lymph organ-derived leukocytes upon antigen exposure in vitro. However, it did not lead to an increase in CD4+CD25+FoxP3+ T_regs_ cells.

Phosphorylcholine (PC) is a component of both oxLDL and multiple prokaryotic organisms, including *Streptococcus pneumonia*, and the formation of autoantibodies against PC after vaccination with the pneumococcal polysaccharide may contribute to reduced atherosclerosis and related cardiovascular events [143]. While passive immunization with monoclonal IgM antibodies against PC reduces atherosclerosis in mice, the transfer of CD11c+ DCs pulsed with PC-keyhole limpet hemocyanin significantly increases atherosclerotic lesion size when injected into *Apoe*−/− mice [142,144]. This effect is associated with increased CD4+ T cell and CD68+ macrophage infiltration within atherosclerotic lesions. Pulsed DCs exhibited an upregulation of maturation markers, promoted the secretion of IFN-γ and IL-17a in T cells, and were associated with increased T_H_1 and T_H_17 populations in the spleen. In addition, PC and oxLDL-specific IgG2a antibodies were increased after the transfer of pulsed DCs.

Depending on the antigen, DC pulsing may evolve as a powerful approach to modulate atherosclerosis. Further research to understand the underlying molecular mechanisms is, however, crucial to promote atheroprotective and to circumvent proatherogenic effects.

#### 3.4.2. Pharmacological Modulation of Dendritic Cells

While genetic perturbation of DC function and vaccination with pulsed DCs are valuable tools for studying the effects on atherosclerosis development and underlying molecular mechanisms, their translation into human studies and clinical practice presents challenges. An alternative approach to accelerate clinical translation involves repurposing easily accessible and affordable small molecules that can modulate DC function and potentially impact atherosclerosis. Vitamin D/dexamethasone, colchicine, and atorvastatin have shown initial results in modulating DC function or abundance in either mice or humans [145,146,147].

In *Apoe–/–* male mice, subcutaneous administration of low-dose vitamin D and dexamethasone resulted in increased IL-10 expression in CD11c+ MHC-II+ cells, as well as other APCs, and induced the generation of Tr1-like T_reg_ cells [145]. Consequently, the treatment led to a reduction in the development of atherosclerotic lesions without causing systemic toxicity or relevant immunosuppression.

Colchicine, an anti-inflammatory drug that inhibits microtubule formation, has demonstrated its potential in reducing the risk of recurrent ischemic cardiovascular events in patients with recent myocardial infarction [12]. In addition to its numerous effects, patients undergoing three months of low-dose colchicine treatment experienced a significant increase in DCs in the circulation [146]. Lastly, a four-week treatment with atorvastatin prior to abdominal aorta aneurysm replacement in humans showed a reduction in DCs within the aneurysmal wall compared to patients who did not receive atorvastatin treatment [147].

## 4. Conclusions

Our understanding of the role of DCs in atherosclerosis has evolved considerably over the last few decades. Starting from the identification of DCs in healthy vasculature and atherosclerotic lesions, DCs are now perceived as central orchestrators of adaptive immunity in atherosclerosis. The advent of omics-based technologies has enabled more accurate classification of DCs and their subtypes and has helped to elucidate their function in more detail. Depending on the subset, DCs can exacerbate or mitigate the inflammatory response to antigens. Understanding the distinct roles of each DC subset and unraveling the underlying molecular mechanisms involved are ongoing areas of research.

Although the therapeutic potential of DC-based interventions in atherosclerosis is still being explored, preclinical data provide support for the idea that targeting DCs could open up new avenues for immune modulation, induction of immune tolerance, and potentially slowing the progression of atherosclerosis. Further research is needed to fully comprehend the intricacies of DCs and to optimize the utilization of DC-based therapies in the treatment or prevention of atherosclerosis.

## Figures and Tables

**Figure 1 ijms-25-02450-f001:**
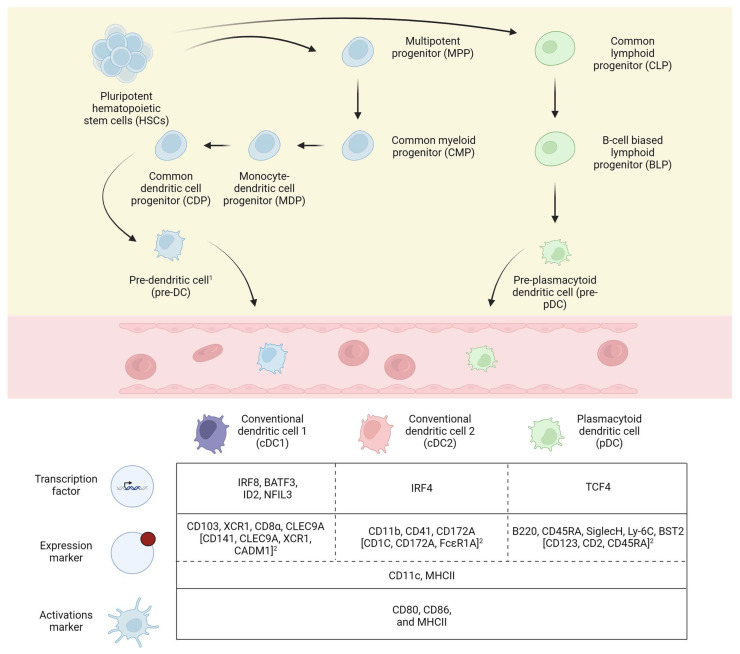
Development of conventional dendritic cells and plasmacytoid dendritic cells from pluripotent hematopoietic stell cells in mice and summary of transcription factors, expression markers and activation markers in conventional dendritic cells 1 and 2 and plasmacytoid dendritic cells according to [42]. BATF3, basic leucine zipper ATF-like transcription factor 3. BST2, bone marrow stromal cell antigen 2. CADM1, cell adhesion molecule 1. CD, cluster of differentiation. ID2, inhibitor of DNA binding 2. IRF8, interferon regulatory factor 8. Ly-6C, lymphocyte antigen 6 family member G6C. NFIL3, nuclear factor, interleukin 3 regulated. TCF4, transcription factor 4. XCR1, X-C motif chemokine receptor 1. CLEC9A, C-type lectin domain containing 9A. ^1^ Pre-dendritic cells can give rise to conventional dendritic cells 1 and 2 (pre-cDCs) and plasmacytoid dendritic cells (pre-pDCs). ^2^ Expression markers used to identify and differentiate dendritic cells in humans.

**Figure 2 ijms-25-02450-f002:**
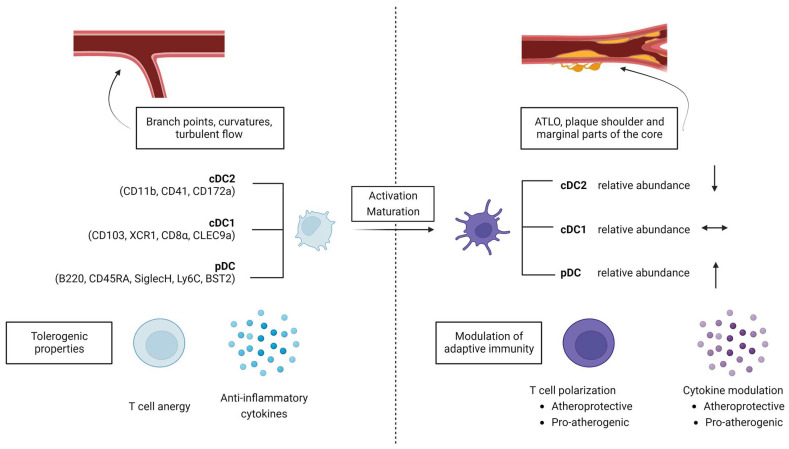
Location, expression profile and properties of dendritic cells in the development of atherosclerosis. ATLO, adventitial artery tertiary lymphoid organs. BST2, bone marrow stromal cell antigen 2. cDC, conventional dendritic cells. Clec4a4, c-type lectin domain family 4, member a4. Ly6C, lymphocyte antigen 6 family member C1. XCR1, X-C motif chemokine receptor 1. pDC, plasmacytoid dendritic cells. Changes in DC subtypes in atherosclerosis according to [72].

**Table 1 ijms-25-02450-t001:** T cells in atherosclerosis [1,20,21,22,23,24,25,26].

T Cell Subtype	Marker	Cytokines Involved in Differentiation/Activation	Subset	Net Effect on Atherosclerosis	Secreted Cytokines
Helper T cell	CD4+	IFN-y, IL-12	T_H_1	Pro-atherogenic	IFN-y, IL-2, IL-3, TNF
		IL-4, IL-5, IL-25, IL-33	T_H_2	Controversial	IL-4, IL-5, IL-10, IL-13
		IL-6, TGF-β, IL-23	T_H_17	Pro-atherogenic	IL-17
		IL-6, IL-2	T_FH_	Pro-atherogenic	IL-21
Cytotoxic T cell	CD8+	IL-12		Controversial	Perforin, granzyme, IFN-γ, TNF-α
Regulatory T cell	CD4+CD25+FoxP3+	IL-2, TGF-β	T_reg_	Atheroprotective	IL-10, TGF-β
	CD4+CD25-FoxP3-	IL-2, IL-6, IL-23, IL-33	exT_reg_	Pro-atherogenic	IL-17, IFN-y
Non-conventional		IL-12, IL-18	iNKT	Pro-atherogenic	T_H_ cytokines
		Il-7	γδ	Pro-atherogenic	IL-17, IFN-y

CD, cluster of differentiation. IFN, interferon. IL, interleukin. iNKT, invariant natural killer T cell. T_FH_, follicular helper cell. TGF, transforming growth factor. TNF, tumor necrosis factor. T_reg_, regulatory T cell. γδ, gamma delta T cell.

**Table 2 ijms-25-02450-t002:** Effects of interferons, interleukins, chemokines, and chemokine receptors on atherosclerosis development in mice [1,30].

	Name	Study Results (Mice)	Net Effect on Atherosclerosis
Interferons			
	IFN-α	Administration accelerates atherosclerosis.	Pro-atherogenic
	IFN-y	Deficiency attenuates atherosclerosis. Administration promotes atherosclerosis.	Pro-atherogenic
	TGF-β	Gene therapy and overexpression reduces atherosclerosis. Inhibition of TGF-β signaling promotes atherosclerosis.	Atheroprotective
	TNF-α	Deficiency reduces atherosclerosis.	Pro-atherogenic
Interleukins			
	IL-1β	Deficiency reduces atherosclerosis.	Pro-atherogenic
	IL-2	Administration of IL-2 enhances atherosclerosis, anti-IL2 AB reduce atherosclerosis.	Pro-atherogenic
	IL-4	Deficiency reduces atherosclerosis in one study; exogenous delivery and deficiency has no effect in another study.	Unclear net effect
	IL-10	Deficiency promotes atherosclerosis, overexpression inhibits advanced lesions.	Atheroprotective
	IL-12	Administration accelerates atherosclerosis. Blockade reduces atherosclerosis.	Pro-atherogenic
	IL-13	Administration promotes plaque stability. Deficiency promotes atherosclerosis.	Atheroprotective
	IL-17a	IL-17 receptor deficiency attenuates atherosclerosis in *Ldlr*−/− mice. Deficiency in *Apoe*−/− mice has no effect on plaque burden or promotes formation of vulnerable plaques. Neutralizing AB in *Apoe*−/− mice attenuates atherosclerosis. Administration of IL-17 promotes atherosclerosis in *Apoe*−/− mice.	Unclear net effect
	IL-23	Neutralizing IL-23 AB reduces pro-inflammatory cytokines. No effect on atherosclerosis in *Apoe−/*− and *Ldlr−/*− mice [31,32].	No effect
	IL-33	Administration reduces atherosclerosis. Treatment with soluble decoy receptor that neutralizes IL-33 leads to larger plaques.	Atheroprotective
Chemokines and receptors			
	CCL19/CCL21	Deficiency in both chemokines increases plaque stability.	Pro-atherogenic
	CCR1	Deficiency increases plaque area.	Atheroprotective
	CCR5	Deficiency protects against atherosclerosis.	Pro-Atherogenic
	CCR7	Deficiency in *Apoe*−/− mice regresses atherosclerosis. Deficiency in *Ldlr*−/− mice exacerbates atherosclerosis.	Unclear net effect
	CXCR6	Deficiency decreases plaque formation.	Pro-atherogenic

AB, antibody. Apoe, apolipoprotein E. CCL, chemokine receptor ligand. CCR, C-C motif chemokine receptor. CXCR, CXC-chemokine receptor. IFN, interferon. IL, interleukin. Ldlr, low-density lipoprotein receptor. TGF, transforming growth factor.

**Table 3 ijms-25-02450-t003:** Localization and change in abundance of DCs and DC subtypes in atherosclerosis.

Study (Year)	Species (Model/ Specimen)	DC Identification (DC Subtype)	DC Localization	Change in Abundance (DC Subtype; Comparator)
Galkina et al. (2006) [68]	Mouse (*Apoe*−/−, +/− WD)	CD11c+/I-Ab+	Aorta	3-fold increase ^1^ (*Apoe−/−* WD vs. WT)
Choi et al. (2011) [67]	Mouse (*Flt3*−/−*Ldlr*−/− + HFD)	MHCII+/CD11c+/CD11b−/CD103+ (cDC1) MHCII+/CD11c+/CD11b+ (cDC2)	Aortic valves, aortic sinus, ascending aorta, descending aorta (intima, adventitia)	2-fold increase ^1^ (cDC1; *Ldlr*−/− HFD vs. WT) 7-fold increase ^1^ (cDC2; *Ldlr*−/− HFD vs. WT)
Koltsova et al. (2012) [14]	Mouse (*Apoe–/–xCd11c-YFP+* +/− WD)	MHCII+/CD11b−/CD11c+ (cDC1) MHCII+/CD11b+/CD11c+ (cDC2)	Aortic arch, valve areas; (intima, adventitia)	3-fold increase ^1^ (*Apoe*–/– WD vs. CD) 3-fold increase ^1^ (*Apoe*–/– WD vs. CD)
Cole et al. (2018) [72]	Mouse (*Apoe*−/−, +/− WD)	MHCII+/CD11c+/CD11b−/CD103+/CD172a− (cDC1) MHCII+/CD11c+/CD11b+/CD172a+ (cDC2) SiglecH+/B220+ (pDC)	Aortic arch, ascending aorta, descending aorta	No statistically significant difference ^2^ (cDC1; (*Apoe*−/− WD vs. CD) Ca 20% decreased ^2^ (cDC2; *Apoe*−/− WD vs. CD) Ca 4-fold increased ^2^ (pDC; *Apoe*−/− WD vs. CD)
Bobryshev and Lord (1995) [58]	Human (thoracic aorta)	Electron microscopy, S-100, CD1a	a. Atherosclerotic lesion resistant region (intima) b. Atherosclerotic lesion predisposed region (intima) c. Atherosclerotic plaque	a. Lowest prevalence b. More prevalent (compared to (a)) c. More prevalent (compared to (a,b))
Yilmaz et al. (2004) [65]	Human (carotid artery)	Fascin, S-100	a. Initial lesion (intima) b. Stable plaque (plaque shoulder, marginal parts of plaque core) c. Vulnerable plaque (plaque shoulder, marginal parts of plaque core)	Significantly increased activated DC population: - 10-fold (advanced plaque vs. initial lesion) - 2-fold (vulnerable vs. stable plaque)
Erbel et al. (2007) [66]	Human (carotid artery, coronary artery)	Fascin, CD83	Shoulder region of unstable plaques	NA
Han et al. (2020) [70]	Human (aortic wall, internal mammary arteries)	Gene expression panel (CIBERSORT algorithm)	NA	Significantly increased activated DC population ^3^ (healthy vs. atherosclerotic vessel, no fold-change reported)
Wang et al. (2021) [69]	Human (carotid, femoral and infra-popliteal peripheral arteries)	Gene expression panel (ImmuCellAI)	NA	Significantly increased DC population from 4.25% to 8.14% ^3^ (healthy vs. atherosclerotic vessel)
Cortenbach et al. (2023) [71]	Human (coronary arteries)	CD1c+/CD20− (cDC2)	NA	Significantly increased absolute cDC2 number (fibrous plaque vs. eccentric intimal thickening, no fold-change reported)

CD, chow diet. cDC1, conventional dendritic cell type 1. cDC2, conventional dendritic cell type 2. DC, dendritic cell. HFD, high-fat diet. NA, not available. pDC, plasmacytoid dendritic cell. WD, Western diet. WT, wild type. YFP, yellow fluorescent protein. ^1^ Absolute difference in DC numbers; ^2^ Relative difference in % of CD45+ cells; ^3^ Relative difference in % of immune cells.

**Table 4 ijms-25-02450-t004:** Overview of DC subtype studies in mice.

	cDC1	cDC2	CCL17	pDC
Study intervention	1. *Flt3*−/− [67] *2. Batf3*−/− [113,115] (BMT [114]) 3. *Irf8^flox/flox^Cd11c^cre^* [116]	1. *Atg16l1^flox/flox^CD11c^Cre^* (BMT) [117] 2. *Clec4a4*−/− [118]	1. CCL17−/− (EGFP TR, BMT, AB) [119]	1. PDCA1 mAB (120G8 [120]) 2 anti–mPDCA-1 [121,122] 3. *Tcf4^−/flox^CD11c^Cre^* (BMT), µMT:pIII+IV^−/−^ [123]
Expression marker	1. CD103+ [67] 2. CD8α+ [113,114]; CD8α+, CD103+ [115] 3. CD8α+, CD103+ [116]	1. CD11b+ [117] 2. CD8α− [118]	1. CD11c+CD11b+ CD8α–CCL17+ [119]	1. PDCA-1+ [120] 2.: Siglec-H+ [121], PDCA-1+ [122] 3. CD11c+ B220+ PDCA1+ [123]
Atherosclerosis model	1. *Ldlr*−/− [67] 2. *Apoe*−/− [113]; *Ldlr*−/− [114,115] 3. *Ldlr*−/− [116]	1. *Ldlr*−/− [117] 2. *Ldlr*−/− [118]	1. *Apoe*−/− [119]	1. *Ldlr*−/− (bilateral semiconstrictive collar placement in the CA) [120] 2. *Ldlr*−/−, *Apoe*−/− [121], *Apoe*−/− [122] 3. *Ldlr*−/− [123]
Plasma lipids	1. ⬌ [67] 2. ⬌ [113,115]; NR [114] 3. TPC ⬆ [116]	1. ⬌ [117] 2. TPC ⬇, TPT ⬇ (male only) [118]	1. ⬌ [119]	1. TPC ⬆ (early in study); TPC ⬌ (during progression) [120] 2. ⬌ [121,122] 3. TPC ⬆ [123]
Study result	1. Atherogenesis ⬆ [67] 2. Atherogenesis ⬇ [113]; Atherogenesis ⬌ [114,115] 3. Atherogenesis ⬇ [116]	1. Atherogenesis ⬇ [117] 2. Atherogenesis ⬇ [118]	1. Atherogenesis ⬇ [119]	1. Atherogenesis ⬆ [120] 2. Atherogenesis ⬇ [121,122] 3. Atherogenesis ⬇ [123]
Suggested mechanism that supported the study results	1. T_regs_ ⬇ [67] 2. T_H_1 response ⬆ [113]; NA [114,115] 3. T_regs_ ⬇, germinal center response ⬇, CD4+ and CD8+ ⬇ [116]	1. Autophagic flux ⬇, T_regs_ ⬆ [117] 2. NR [118]	1. T_regs_ ⬆ [119]	1. IDO expression ⬆; CD4+ T-cell ⬆ [120] 2. Cramp/DNA complex mediated activation of pDCs ⬇, pDC IFN-α expression ⬇ [121], T-cell activation ⬇, pro-atherogenic cytokines ⬇ [122] 3. IFN-γ ⬇, lesional T cell infiltration ⬇ [123]
DC subtype net effect on atherosclerosis development	1. Atheroprotective [67] 2. Pro-atherogenic [113]; No effect [114,115] 3. Pro-atherogenic [116]	1. Pro-atherogenic [117] 2. Pro-atherogenic [118]	1. Pro-atherogenic [119]	1. Atheroprotective [120] 2. Pro-atherogenic [121,122] 3. Pro-atherogenic [123]

AB, antibody. ApoE, apolipoprotein E. Atg16l, autophagy-related 16-like 1. Batf3, basic leucine zipper transcription factor ATF-like 3. BMT, bone marrow transplant. CCL17, chemokine receptor ligand 17. cDC, conventional dendritic cell. Clec4a4, c-type lectin domain family 4, member a4. EGFP TR, targeted replacement of the Ccl17 gene by enhanced green fluorescent protein gene. FLT3, cytokine fms-like tyrosine kinase 3. IDO, 2,3-dioxygenase. IFN-γ, interferon γ. Irf8, interferon regulatory factor 8. Ldlr, low-density lipoprotein receptor. NR, not reported. pDC, plasmacytoid dendritic cell. PDCA-1, plasmacytoid dendritic cell antigen-1. Tcf4, transcription factor 4. T_H_1, T helper 1 cells. TPC, total plasma cholesterol. TPT, total plasma triglycerides. T_regs_, T regulatory cells. ⬆ indicates an increase or higher values compared to controls; ⬇ indicates a decrease or lower values compared to controls; ⬌ indicates no significant changes.

## Data Availability

Not applicable.

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
