# Peer review of "The Evolving Role of Dendritic Cells in Atherosclerosis"

_ijms, 2024, doi:10.3390/ijms25042450_

Round 1

Reviewer 1 Report

Comments and Suggestions for Authors

Dear editor, I read with interest and attention the review article of the review with title “The evolving role of dendritic cells in atherosclerosis” by the authors Simone Britsch , Harald Langer, Daniel Duerschmied , Tobias Becher

The review it is interesting, overall the review is well written, and reads very fluidly.however I have a series of observations about this work that could contribute to improving this review.

Page 2 line 41 delete “adventitial spaces of  the healthy arteries in mice and humans

page 2 line 61 eliminates eliminates the double parenthesis in CD4+))

page 3 line 94 the umlaut in naïve

page 4 line 159 eliminates the double parentheses in CLRs))

page 5 line 178 what does S100 mean?

page 5 line 196 the abbreviation for MHC is already defined previously, please correct

page 5 line 287 the authors could add a heading in this section (The TGF-B)

page 8 line 298 the authors could add a heading in this section (hypoxia-inducible factor)

page 8 line 316 could the authors add a heading in this section (cytokines)

page 8 line 324 change the abbreviation Ido to uppercase IDO

page 9 line 352 the authors could refer to table one with a sentence, the way the mention of the table appears is not acceptable.

In table 1 the authors could add what the arrows mean

page 9 line 404 what does DNGR-1 mean please develop the abbreviation

page 12 line 422 the authors could develop more the section on sexual hormones and the participation of cDC2 because it is very cursory

page 12 line 446 the authors could refer to table one with a sentence, the way the mention of the table appears is not acceptable.

page 15 line 561 the authors could add a reference at the end of the paragraph

The authors refer to figure 1 but the figure does not appear in the text. They could include it in the text and in which section the figure would appear.

The review would benefit from two or three figures in the different sections and not just one figure, which also does not appear in the text.

Since in atherosclerosis both the eNOS and COX 2 and 1 pathway are very importantly involved, the authors could add to the review the participation of both pathways and the association that exists with DCs.

I thank the editor for the opportunity to review this article

Comments on the Quality of English Language

Overall the review is well written, and reads very fluidly

Author Response

Reviewer 1

Dear editor, I read with interest and attention the review article of the review with title “The evolving role of dendritic cells in atherosclerosis” by the authors Simone Britsch, Harald Langer, Daniel Duerschmied, Tobias Becher

The review it is interesting, overall the review is well written, and reads very fluidly. however I have a series of observations about this work that could contribute to improving this review.

Thank you for your positive assessment of our review. Below, you will find our detailed response to your observations and comments, addressed point by point.

Page 2 line 41 delete “adventitial spaces of the healthy arteries in mice and humans

We deleted parts of the sentence as recommended by the reviewer. The sentence now reads:

Dendritic cells (DCs) are a subtype of antigen-presenting cells (APCs) that reside in the intimal and adventitial spaces of arteries.

page 2 line 61 eliminates eliminates the double parenthesis in CD4+))

We reworded the sentence to clarify the abbreviations in parenthesis. The sentence now reads:

T cells can be classified into different subtypes, including helper T cells (TH; cluster of differentiation 4 positive (CD4+)), cytotoxic T cells (CD8+) and regulatory T cells (Treg; CD4+CD25+ and additional subsets).

page 3 line 94 the umlaut in naïve

We removed the umlaut as indicated. The sentence now reads:

In addition, DCs express abundant MHC II proteins on their surface, are potent activators of naive T cells and B cells, and thus play a pivotal role in connecting innate and adaptive immunity [33, 34].

A similar umlaut was corrected on page 2, line 66. The sentence now reads:

After recognition through a T cell receptor, naive CD4+ cells differentiate and migrate to the atherosclerotic lesion, mediated by chemokines and their corresponding homing receptors like CXC-chemokine receptor (CXCR) 3, CXCR6, C-C-Motiv-Chemokin-Rezeptor (CCR) 5 and CCR1 [1, 15, 18].

page 4 line 159 eliminates the double parentheses in CLRs))

In this sentence, the double parenthesis is actually required because it concludes the following part of the sentence in parenthesis:

(PRRs; including toll like receptors (TLR) and c-type lectin reporters (CLRs))

We thus left this sentence unchanged and hope that the reviewer agrees.

page 5 line 178 what does S100 mean?

S-100 refers to the expression of the S-100 protein which was detected by immunostaining in cells with a dendritic shape (Bobryshev YV and Lord RSA, 1995; reference [56])

We reworded the sentence to provide more clarity and removed the two proteins that were stained for (S-100 and CD1a):

DCs have been reported as early as 1995 in ultrathin sections of human aortic intima, collected from trauma victims during autopsy and examined by electron microscopy and immunohistochemistry [56].

page 5 line 196 the abbreviation for MHC is already defined previously, please correct

We corrected the wording as suggested. In addition, dendritic cells was also already defined and we used the abbreviation DC accordingly. The sentence now reads:

In healthy mouse aortas, CD11c+ DCs can effectively cross-present antigens on MHCI to CD8+ T cells, indicating a role in maintaining immunogenic hemostasis [17].

In addition, we corrected the abbreviation DC throughout the manuscript in cases were the term dendritic cells is currently used (except in headings and figure legends to improve readability).

page 5 line 287 the authors could add a heading in this section (The TGF-B)

page 8 line 298 the authors could add a heading in this section (hypoxia-inducible factor)

page 8 line 316 could the authors add a heading in this section (cytokines)

We thank the reviewer for these useful suggestions and agree that the structure of individual sections of the manuscript can be improved by adding appropriate subsection headers. To address the 3 comments above and to provide consistency, we have added the following headers to section 3.3.1. CD11c+ dendritic cells:

3.3.1.1. Krüppel-like factor 2

3.3.1.2. Myeloid differentiation primary response protein 88

3.3.1.3. Human B-cell lymphoma 2

3.3.1.4. TGF-β receptor II signaling

3.3.1.5. Hypoxia-inducible factor

3.3.1.6. Indoleamine 2, 3-dioxygenase

3.3.1.7. Cyclooxygenase 1, 2 and endothelial nitric oxide synthase

For section 3.3.2. Conventional dendritic cell subtypes 1 and 2, we have added the following headers:

3.3.2.1 Fms-like tyrosine kinase 3

3.3.2.2 Transcription factors basic leucine zipper transcription factor ATF-like 3 and Interferon regulatory factor 8

3.3.2.3 Autophagy-related 16 like 1

3.3.2.4 C-type lectin receptors CLEC9a and CLEC4A4

For section 3.3.4. Plasmacytoid dendritic cells, we have added the following headers:

3.3.4.1 Anti-mouse PDCA1 antibodies

page 8 line 324 change the abbreviation Ido to uppercase IDO

We thank the reviewer for the careful observation. We changed the text accordingly:

Stable overexpression of IDO in an immature DC cell line derived from bone marrow progenitor cells and subsequent transfer into the Apoe-/- mouse model reduces atherosclerosis development, independent of changes in lipid metabolism.

page 9 line 352 the authors could refer to table one with a sentence, the way the mention of the table appears is not acceptable.

We agree with the author comment. To properly refer to the table (now table 4), we have included the following sentence in the manuscript:

Table 4 summarizes the different approaches and the observed net effect on atherosclerosis development.

In table 1 the authors could add what the arrows mean

To provide more clarity, the following sentence was added to the table legend (now table 4):

indicates an increase or higher values compared to controls; indicates a decrease or lower values compared to controls; indicates no significant changes.

page 9 line 404 what does DNGR-1 mean please develop the abbreviation

We provide the full name of DNGR-1 to provide more clarity:

Dendritic cell natural killer lectin group receptor-1 (DNGR-1; also known as CLEC9a) is a CLR predominantly expressed on CD8a+ cDC1s, and it acts as a receptor for dead cell-associated antigens, which may accumulate in the necrotic cores of advanced atherosclerotic lesions [120].

page 12 line 422 the authors could develop more the section on sexual hormones and the participation of cDC2 because it is very cursory

We thank the reviewer for the suggestion. To summarize the current evidence on a high level, we have included the following sentences:

Previous studies have implicated estrogen and estrogen receptor alpha (ERα) signal-ing in the development and functional capacities of DCs [122, 123]. ERα, a ligand-dependent transcription factor, is expressed in both progenitor and mature DCs [124]. Its signaling promotes IRF4 expression in monocyte-derived DC pro-genitors and their subsequent differentiation [125]. In the Flt3 ligand-dependent model of DC differentiation, ERα signaling enhances the development of cDCs and pDCs, which exhibit more pronounced pro-inflammatory cytokine production in response to TLR stimulation. The overall effect of estrogen and ERα signaling on cDC2s in particular is currently unclear.  Further investigation is also needed to understand the precise role of CLEC4A4 in cDC2s and the molecular mechanisms underlying its suggested pro-atherogenic effect.

page 12 line 446 the authors could refer to table one with a sentence, the way the mention of the table appears is not acceptable.

We agree that table 1 (now table 4) is not adequately introduced in this sentence. We have removed the reference to the table from this section of the manuscript as it is a) introduced more adequately earlier in the manuscript and b) the information provided is succinctly captured in the main text.

page 15 line 561 the authors could add a reference at the end of the paragraph

To address the reviewer’s comment, we moved reference [114] to the previous sentence. We also included the 3 references [114, 115, 116] at the end of the first paragraph to reference the source of the statement. Please note that the number of the references changes due to updates:

The changes to the manuscript are as follows:

Vitamin D/dexamethasone, colchicine, and atorvastatin have shown initial results in modulating DC function or abundance in either mice or humans [143-145].

In Apoe–/– male mice, subcutaneous administration of low-dose vitamin D and dexamethasone resulted in increased IL-10 expression in CD11c+ MHC-II+ cells, as well as other APCs, and induced the generation of Tr1-like Treg cells [143].

The authors refer to figure 1 but the figure does not appear in the text. They could include it in the text and in which section the figure would appear.

We agree with the reviewer that it helps to place Figure 1 to see it in context. We envision Figure 1 and the corresponding legend to appear immediately after the following sentence in the manuscript:

            The data presented provides compelling evidence for the involvement of the overall CD11c+      DC population in modulating atherosclerosis. It is important to note that different subtypes of     DCs may have distinct contributions to plaque development, each potentially contributing to      net pro-atherogenic or atheroprotective effects (Figure 1).

The review would benefit from two or three figures in the different sections and not just one figure, which also does not appear in the text.

We agree with the reviewers´ comment and have added the following tables and figures:

Table 1 T cells in atherosclerosis.

Table 2 Effects of interferons, interleukins, chemokines and chemokine receptors on atherosclerosis development in mice.

Table 3 Localization and change in abundance of DCs and DC subtypes in atherosclerosis.

Figure 1 Development of conventional dendritic cells and plasmacytoid dendritic cells from pluripotent hemato-poietic stell cells in mice and summary of transcription factors, expression markers and activation markers in con-ventional dendritic cells 1 and 2 and plasmacytoid dendritic cells according to [40]

Since in atherosclerosis both the eNOS and COX 2 and 1 pathway are very importantly involved, the authors could add to the review the participation of both pathways and the association that exists with DCs.

We agree with the reviewer that COX1/2 and eNOS are important pathways to consider and have thus added the following text:

  1. 3.1.7. Cyclooxygenase 1, 2 and endothelial nitric oxide synthase

Cyclooxygenase (COX) is the rate-limiting enzyme in the synthesis of prostaglandins (PGs) from arachidonic acid. The COX gene encodes two isoenzymes called COX-1 and COX-2. While COX-1 is constitutively expressed, COX-2 is induced rapidly in response to inflammation-causing cytokines (94). In the context of atherosclerosis, COX-2 expression is upregulated in endothelial cells, smooth muscle cells and macrophages, and it accumulates in atherosclerotic lesions (95, 96). PGs have a significant impact on atherosclerosis and cardiovascular health and can exert atheroprotective (e.g. PGI2) and proatherogenic effects (e.g. PGE2).

COX-2 expression and PGE2 production is increased in DCs after LPS exposure (97, 98). COX-2 derived PGE2 has been linked to the survival of DC pro-genitors and the Flt3 ligand-dependent development of CD11c+CD11b+ cDCs (but not pDCs) (99). Moreover, PGE2 signaling modulates DC migration and maturation via E-prostanoid 2/E-prostanoid 4 (EP2/EP4) receptors (100-102). It also affects the release of IL-10, IL-12 and IL-23 by DCs (97). In a mouse model of type 1 diabetes related accelerated atherogenesis however, myeloid cell-targeted deletion of the EP4 receptor impaired PGE2 signaling and altered cytokine production in DCs but did not modulate atherosclerosis development (103).

Nitric oxide (NO) is generated by one of the three NO synthase (NOS) isoforms called neuronal (nNOS; NOS3), inducible (iNOS; NOS2) and endothelial (eNOS; NOS1) NOS (104). These NOS enzymes facilitate the conversion of L-arginine, NADPH, and O2 into L-citrulline, NADP+, and NO. NO, a gas that can permeate membranes, plays a crucial role in modulating vascular tone, endothelial function, leucocyte chemotaxis, and platelet adhesion, and the disturbances in the equilibrium of NO are significant contributors to atherosclerosis. In murine monocyte-derived DCs, iNOS expression has been observed and can be induced by IFN-γ or LPS (105, 106). NO triggers metabolic reprogramming in DCs, characterized by sustained glycolysis and the suppression of mitochondrial activity, which is mediated by mTOR/HIF1α/iNOS (107). These metabolic changes are associated with in-creased production of inflammatory cytokines, upregulation of costimulatory mole-cules, enhanced T cell stimulatory capacity, and prolonged DC survival (106, 108). Although iNOS has been detected in DCs present in human carotid artery atherosclerosis, the specific effects of NO on DCs in this context remain unclear (109).

Reviewer 2 Report

Comments and Suggestions for Authors

In the presented paper, authors are describing role of dendritic cells and DCs subpopulations in the atherosclerosis – a relevant aspect in the field. DCs as antigen presenting cells recruit other immune cells and stimulate them. Depending on subtype of DCs they can play pro- and anti – atherosclerosis properties.

Though potentially interesting, the paper is overall badly written, which is a pity. The text is written in such an overwhelming way that the reader is continuously lost. The continuous text trying to cover too many topics, all intertwined. As said, that's a real pity, because the important impact of DCs on atherosclerosis.

Lines 58-59: Its main effectors are T cells, APCs and cytokines [13]. – the order in followed text is different – please change accordingly.

Line 62: and additional subsets. – please specify.

Lines 69-70: Depending on their subtype, T cells may mediate atheroprotective (e.g. Treg cells) or pro-atherogenic effects (e.g. TH1) [1, 18, 19] – what cytokines recruit what subtype of T cells?

2. Cellular immunity and atherosclerosis

For both, T cells and cytokines mentioned in this paragraph make tables and point their properties, markers. It will be simpler to present.

Line 83: APCs (including dendritic cells, macrophages, and B cells) – please point out all APCs.

Lines 99-172: Graphical presentation how DCs are created and what subtypes we can recognize, and comparison between species (human vs. mouse) would be preferred as figures. The plain text is too dense. 

Lines 199-200: Compared to the low numbers of DCs in healthy vasculature, DCs increase in abundance during plaque development in mice and humans [14, 47, 54-60]. – please make a table to support this data. If the authors performed SCs classification – also include the data.

3.4.2. Pharmacological modulation of dendritic cells in human studies

Lines 569-570 Limited data in both mice and humans are available for three such molecules: what datasets were accessed and at what day?

Lines 572-573 In Apoe–/– male mice, subcutaneous administration of low-dose vitamin D and dexamethasone – study performed on mice and the title of the paper point human study. Please change.

Author Response

Reviewer 2

In the presented paper, authors are describing role of dendritic cells and DCs subpopulations in the atherosclerosis – a relevant aspect in the field. DCs as antigen presenting cells recruit other immune cells and stimulate them. Depending on subtype of DCs they can play pro- and anti – atherosclerosis properties.

Though potentially interesting, the paper is overall badly written, which is a pity. The text is written in such an overwhelming way that the reader is continuously lost. The continuous text trying to cover too many topics, all intertwined. As said, that's a real pity, because the important impact of DCs on atherosclerosis.

We genuinely appreciate the valuable input provided by the reviewer on our manuscript and the insightful suggestions for improvement. We fully acknowledge the significance of the topic and deeply regret any concerns raised regarding the content and structure. By addressing your comments, along with those of reviewer 1, we aim to make significant improvements to the manuscript. Please find below a detailed point-by-point response to each observation and comment provided.

Lines 58-59: Its main effectors are T cells, APCs and cytokines [13]. – the order in followed text is different – please change accordingly.

We thank the reviewer for the careful observation. We changed the text accordingly to reflect the order in which the T cells, cytokines and APCs in the paragraph below:

Its main effectors are T cells, cytokines and APCs [13].

Line 62: and additional subsets. – please specify.

We are providing additional context regarding T cells (including Tregs) as part of Table 1 in the manuscript. We thus reworded the sentence as follows and refer to Table 1 with more details in the paragraph below:

T cells can be classified into different subtypes, including helper T cells (TH; cluster of differentiation 4 positive (CD4+)), cytotoxic T cells (CD8+) and regulatory T cells (Treg; CD4+CD25+Foxp3+).

Lines 69-70: Depending on their subtype, T cells may mediate atheroprotective (e.g. Treg cells) or pro-atherogenic effects (e.g. TH1) [1, 18, 19] – what cytokines recruit what subtype of T cells?

To provide this information, we added Table 1 T cells in atherosclerosis to the manuscript.

  1. Cellular immunity and atherosclerosis

For both, T cells and cytokines mentioned in this paragraph make tables and point their properties, markers. It will be simpler to present.

We thank the reviewer for the valuable suggestion. We have added the following tables to the manuscript:

Table 1 T cells in atherosclerosis.

Table 2 Effects of interferons, interleukins, chemokines and chemokine receptors on atherosclerosis development in mice.

Line 83: APCs (including dendritic cells, macrophages, and B cells) – please point out all APCs.

According to the reviewer´s suggestion, we have extended this section to include professional and non-professional APCs:

Lastly, antigen-presenting cells (APCs) are responsible for capturing and processing antigens. Both modified endogenous antigens and, to a lesser extent, exogenous antigens have been implicated in atherosclerosis [31]. APCs can be broadly differentiated into professional APCs (such as dendritic cells, macrophages, and B cells) and non-professional APCs. The defining features of professional APCs include their ability to acquire exogenous antigens and present antigen peptides via MHCII, express co-stimulatory molecules, and secrete cytokines. These features enable professional APCs to train and activate CD4+ T cells. On the other hand, non-professional APCs possess the ability to present peptides derived from endogenous proteins via MHCI to CD8+ cytotoxic T cells, and all nucleated cells can be considered non-professional APCs.

Lines 99-172: Graphical presentation how DCs are created and what subtypes we can recognize, and comparison between species (human vs. mouse) would be preferred as figures. The plain text is too dense. 

We agree with the reviewer’s suggestion and have added the following figure to the manuscript:

Figure 1 Development of conventional dendritic cells and plasmacytoid dendritic cells from pluripotent hemato-poietic stell cells in mice and summary of transcription factors, expression markers and activation markers in con-ventional dendritic cells 1 and 2 and plasmacytoid dendritic cells according to [40]

Lines 199-200: Compared to the low numbers of DCs in healthy vasculature, DCs increase in abundance during plaque development in mice and humans [14, 47, 54-60]. – please make a table to support this data. If the authors performed SCs classification – also include the data.

We thank the reviewer for the suggestion. We added Table 3 Localization and change in abundance of DCs and DC subtypes in atherosclerosis which depicts the changes in DCs and subtypes in mice and humans during plaque development.

3.4.2. Pharmacological modulation of dendritic cells in human studies

Lines 569-570 Limited data in both mice and humans are available for three such molecules: what datasets were accessed and at what day?

We agree with the reviewer that this statement may imply a systematic, unbiased search across different datasets. To specify, we reworded the sentence accordingly:

            Vitamin D/dexamethasone, colchicine, and atorvastatin have shown initial results in modulating DC function or abundance in either mice or humans [143-145].

Lines 572-573 In Apoe–/– male mice, subcutaneous administration of low-dose vitamin D and dexamethasone – study performed on mice and the title of the paper point human study. Please change.

We thank the reviewer for the careful observation. We changed heading 3.4.2 accordingly, which now reads:

            3.4.2 Pharmacological modulation of dendritic cells 
